# KeFVP: Knowledge-enhanced Financial Volatility Prediction

**Hao Niu[1], Yun Xiong[1]\*Xiaosu Wang[1], Wenjing Yu[1], Yao Zhang[1],**
**Weizu Yang[2]**

[1]Shanghai Key Laboratory of Data Science, School of Computer Science, Fudan University
[2]Shanghai Yuanlu Jiajia Information and Technology Co., Ltd.
[1]{hniu18, yunx, xswang19, yaozhang}@fudan.edu.cn
[1]{wjyu21}@m.fudan.edu.cn
[2]{weizu.yang}@jiajiagroup.net

## Abstract

Financial volatility prediction is vital for characterizing a company's risk profile. Transcripts of companies' earnings calls serve as valuable, yet unstructured, data sources to be utilized to access companies' performance and risk profiles. Despite their importance, current works ignore the role of financial metrics knowledge (such as EBIT, EPS, and ROI) in transcripts, which is crucial for understanding companies' performance, and little consideration is given to integrating text and price information. In this work, we statistically analyze common financial metrics and create a special dataset centered on these metrics. Then, we propose a **k**nowledge-**e**nhanced **f**inancial **v**olatility **p**rediction method (KeFVP) to inject knowledge of financial metrics into text comprehension by **k**nowledge-**e**nhanced adaptive **p**re-**t**raining (KePt) and effectively integrating text and price information by introducing a conditional time series prediction module. Extensive experiments are conducted on three real-world public datasets, and the results indicate that KeFVP is effective and outperforms all the state-of-the-art methods. [1]

## 1 Introduction

The volatility of financial asset prices is typically considered a valid proxy for the risk of financial assets and plays an essential role in evaluating the risk of financial assets and their derivatives (Yang et al., 2020). Predicting the volatility of financial assets is therefore of great significance to market participants. Meanwhile, in addition to asset price information, a wealth of unstructured data (e.g., news, social media, etc.) (Ding et al., 2014, 2015; Xu and Cohen, 2018; Duan et al., 2018; Yang et al., 2019; Feng et al., 2019) can also reflect potential changes in the future volatility of assets, which is also vital information that the market participants

should be aware of. One such unstructured data source is earnings calls, which are quarterly conferences held by public company management to explain the latest performance, offer guidance on their expectation for the coming future, and answer questions raised by investors and analysts (Qin and Yang, 2019). The information conveyed by the conference provides investors and analysts with valuable insights into the company's current state and future prospects. Hence, the goal of this task is to predict future stock price volatility after the earnings call announcement by combining historical prices and earnings call transcripts.

Recent works (Qin and Yang, 2019; Sawhney et al., 2020c; Yang et al., 2020; Sawhney et al., 2020b) have also embarked on exploring the approaches of utilizing these earnings calls to improve financial volatility predictions. Prior works based on earnings calls pay attention to multi-task architecture for predicting volatility and price movement (Sawhney et al., 2020c; Yang et al., 2020), correlations between stocks (Sawhney et al., 2020b), and the impact of numeric features (Yang et al., 2022). However, these studies either completely overlooked the role of price (Yang et al., 2020, 2022; Qin and Yang, 2019), or did not consider the combination of price and text information elaborately (Sawhney et al., 2020b,c). For instance, VolTAGE (Sawhney et al., 2020b) encodes price information by a vanilla LSTM, while Ensemble (Sawhney et al., 2020c) uses Support Vector Regression to predict volatility based on historical price information. Thus, based on the preceding review, we propose two existential challenges in existing studies: (1) financial metric (FM) knowledge is not concerned; (2) combining text and price information is rarely considered elaborately.

Specifically, there are a large number of FMs (e.g., EPS, EBITDA[2], etc. as shown in Table 1)

---

\*Corresponding author
[1]The code is at https://github.com/hankniu01/KeFVP

[2]Please refer to Table 9 in Appendix B for full names and descriptions

Table 1: Statistics on datasets. *#Ave. Sent.* denotes the average number of sentences in each transcript. The column *#Total TS* gives the overall number of transcripts. *Total Sent.* denotes the overall count of sentences, while *FM Sent.* denotes the number of sentences containing FM. And *FM Ratio* is the ratio of the previous two.

| Dataset | | #Ave. Sent. | #Total TS | #Total Sent. | #FM Sent. | FM Ratio (%) |
|---|---|---|---|---|---|---|
| EC | Train | 157 | 391 | 61434 | 38150 | **62.10** |
| | Test | 157 | 112 | 17589 | 10576 | **60.13** |
| | Dev. | 160 | 56 | 8940 | 5594 | **62.57** |
| | Overall | 156 | 559 | 87963 | 54320 | **61.75** |
| MAEC-15 | Train | 95 | 535 | 50632 | 25542 | **50.45** |
| | Test | 92 | 154 | 14234 | 7387 | **51.90** |
| | Dev. | 100 | 76 | 7571 | 3609 | **47.67** |
| | Overall | 95 | 765 | 72437 | 36538 | **50.44** |
| MAEC-16 | Train | 101 | 980 | 99246 | 51622 | **52.01** |
| | Test | 84 | 280 | 23557 | 11436 | **48.55** |
| | Dev. | 100 | 140 | 14058 | 7597 | **54.04** |
| | Overall | 98 | 1400 | 136861 | 70655 | **51.63** |

in the financial text to describe companies' performance in terms of earnings, cash flow, and assets and liabilities, which offer important assistance in analyzing companies' performance. However, to the best of our knowledge, there is rarely work that aims to use such FM knowledge to enhance financial predictions. Secondly, the Efficient Market Hypothesis (EMH) (Malkiel, 1989; Sawhney et al., 2020a) suggests that financial markets are informationally efficient, meaning that stock prices reflect all available market information. Consequently, in addition to historical prices, text information also affects stock prices. Thus, the concurrent integration of price and text is of paramount importance.

In this work, we propose a novel approach, **k**nowledge-**e**nhanced **f**inancial **v**olatility **p**rediction (KeFVP), to tackle the challenges mentioned above. The overview of KeFVP is shown in Figure 1. Initially, we introduce a **k**nowledge-**e**nhanced adaptive **p**re-**t**raining (KePt) method, designed to inject FM knowledge into Pre-trained Language Models (PLMs). To facilitate this, we construct a KePt dataset that merges financial corpora (e.g., TRC2-financial[3] and FiQA[4], etc.) and extracts specific descriptions of FMs from Wikidata[5] for use in the KePt process. Subsequently, we employ the PLM post-KePt to extract representations of each sentence from the earnings call transcripts. These representations are then incorporated into an end-to-end **f**inancial **v**olatility **p**rediction (FVP) model along with historical prices for volatility prediction. The FVP model consists of two major components: an information enhancement (IE) module and a conditional time series prediction (CTSP) module.

[3] https://trec.nist.gov/data/reuters/reuters.html
[4] https://sites.google.com/view/fiqa/home
[5] https://www.wikidata.org/wiki/Wikidata:Main_Page

Firstly, the text information is directed into the IE module, composed of multiple Transformer blocks. Following the IE module's processing, the refined text information, along with historical prices, is fed into the CTSP module to carry out volatility predictions. Our main contributions are as follows:

- We first highlight the overlooked issue of disregarding FMs in existing studies. To counteract this challenge, we develop a knowledge-enhanced adaptive pre-training (KePt) method to inject FMs knowledge into PLMs and construct a specific KePt dataset centered on FMs for adaptive pre-training.

- We proposed an FVP model, equipped with IE and CTSP modules, designed to effectively amalgamate price and text information.

- We perform evaluations using three real-world earnings call datasets, and our results establish new state-of-the-art (SOTA) benchmarks.

## 2 Related Work

### 2.1 Integrate Financial Knowledge into PLMs

In finance, financial metrics (FMs) serve as crucial indicators of understanding companies' performance, financial, and operating status when executives or analysts read financial texts. Nevertheless, vanilla PLMs ignore the processing of FMs, and few researchers have recognized such a challenge. Meanwhile, in the general field, injecting knowledge into PLMs during pre-training has been investigated (Yu et al., 2022; Sun et al., 2020; Wang et al., 2021a) to some extent. Hence, we extract FM descriptions from Wikidata as knowledge and propose the KePt method to infuse such financial knowledge into PLMs during pre-training. To our best knowledge, it is the first attempt to focus on FM knowledge in the processing of financial texts.

### 2.2 Earnings Call Data

Earnings calls present explanations of companies' performance, guidance for the upcoming quarter, and opportunities for in-depth Q&A, which provides a good window of communication between investors, brokerage analysts, and company management (Keith and Stent, 2019). The earnings call datasets we used were released by (Qin and Yang, 2019; Li et al., 2020). Existing studies primarily focus on exploring three main aspects: (a) multimodal fusion, some works (Qin and Yang,

2019; Sawhney et al., 2020c; Yang et al., 2020) are dedicated to exploring the combination of text and audio modalities and its benefits for volatility predictions; (b) inter-company relationship, such as VolTAGE (Sawhney et al., 2020b), which utilizes graph neural networks to incorporate stock interdependence into prediction; and (c) the characteristics of financial texts, for instance, NumHTML (Yang et al., 2022) explores the importance of numeric structure conveyed by numbers in texts for financial prediction. However, these works do not integrate price and text information elaborately, nor do they recognize the significance of FMs in texts.

## 2.3 Financial Prediction with Text

Financial predictions have been greatly enhanced by the incorporation of text information, such as financial news and social media. Current research can be grouped into four distinct categories. (1) Event-based Prediction. These approaches leverage event information extracted from financial news to guide financial predictions. Significant works in this domain include (Ding et al., 2014, 2015) and more recent developments (Yang et al., 2019; Deng et al., 2019). (2) Plain Text-based Prediction. Such methods (Duan et al., 2018; Xu and Cohen, 2018) involve learning directly from unstructured data such as tweets or news documents, without pre-extracting structured events. (3) Inter-company Relationships. This genre of studies (Ang and Lim, 2022; Sawhney et al., 2020a; Xu et al., 2021; Cheng and Li, 2021) takes into account the inter-company relationships while considering text information. (4) Portfolio Management. This type of works (Liang et al., 2021; Sawhney et al., 2021a; Du and Tanaka-Ishii, 2020; Sawhney et al., 2021b) targets portfolio management problems instead of financial predictions by exploiting textual information.

## 2.4 Stock Market Volatility Prediction

In the stock market, volatility prediction plays a central role in risk management, asset allocation, and derivative pricing (Liang et al., 2022; Ma et al., 2019; Bollerslev et al., 2009; Epstein and Ji, 2013). In the field of finance, research on volatility predictions primarily focuses on two aspects. Firstly, works in this aspect aim to construct forecasting methods based on widely used financial models such as GARCH and ARIMA (Dai et al., 2022; Spyridon D. Vrontos and Vrontos, 2021; Wang et al., 2016; Engle and Patton, 2007). The second aspect is innovation on the data side (None-

jad, 2017; Zhang et al., 2022; Audrino et al., 2020; Chen et al., 2020; Wang et al., 2021b). For instance, (Nonejad, 2017; Zhang et al., 2022) incorporates macroeconomic indicators into the volatility predictions; and (Audrino et al., 2020) finds that analyzing market emotions can enhance the effectiveness of volatility prediction.

## 3 Approach

The overview of KeFVP is shown in Figure 1, which is made up of two components: (1) **k**nowledge-**e**nhanced adaptive **p**re-**t**raining (KePt) (top); and a (2) **f**inancial **v**olatility **p**rediction (FVP) model (bottom). The FVP model consists of two major modules: the (i.) information enhancement (IE) module (bottom(a)); the (ii.) conditional time series prediction (CTSP) module (bottom(b)).

## 3.1 Knowledge-enhanced Adaptive Pre-training (KePt)

We introduce the KePt method and the KePt dataset we constructed to answer the challenge of disregarding financial metric (FM) knowledge.

### 3.1.1 KePt Dataset

To construct the KePt dataset, we collect Financial PhraseBank [6], FiQA (both Task1 and Task 2), and EC dataset as financial corpora. Then, we extract descriptions of frequent FMs from Wikidata. The key FMs, along with their descriptions and the frequency of their appearances across earnings call datasets and the KePt dataset, are displayed in Table 9 in Appendix B. Next, guided by our precompiled list of frequent FMs, we sift through the financial corpora to isolate sentences that include these metrics. These sentences are subsequently integrated into the KePt dataset. Accompanying descriptions relevant to these FMs are also incorporated. A comprehensive statistic is provided in Table 8 in Appendix B. Please note that each FM is accompanied by a corresponding description, and every sentence comprises at least one FM. To foster future research endeavors, we will make the KePt dataset publicly available with our source code.

### 3.1.2 KePt Method

The KePt method is illustrated in Figure 1(top). We commence with the published BERT$_{BASE}$ (bert-base-uncased[7]) as our foundation and proceed to

---

[6]https://www.researchgate.net/publication/251231364_FinancialPhraseBank-v10

[7]https://huggingface.co/bert-base-uncased

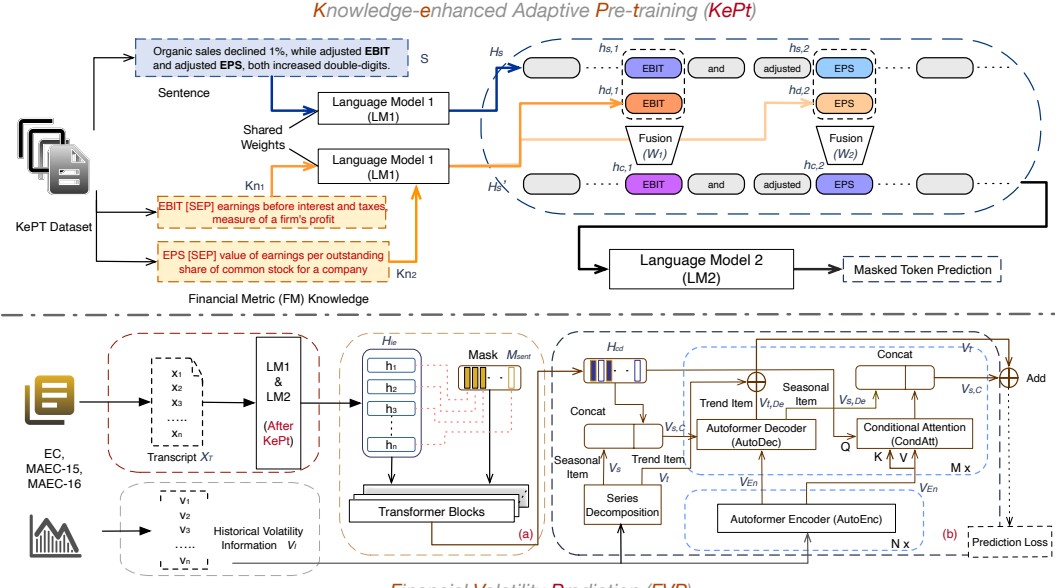

Figure 1: The overall architecture of KeFVP. KePt at the top, FVP at the bottom.

conduct training using the KePt method. We designate the embedding layer and the initial 6 encoders of the BERT$_{\text{BASE}}$ as Language Model 1 (LM1) and assign the remaining layers to Language Model 2 (LM2). LM1, which is used to encode both sentences and FMs, shares its weight parameters.

Concretely, the input sentence is represented as $S = \langle [\text{CLS}], w_1, \cdots, [\text{MASK}] \cdots, w_N, [\text{SEP}] \rangle$, while FMs in this sentence, and their descriptions, are represented as $Kn = \langle [\text{CLS}], SP_{fm}, [\text{SEP}], SP_{desc}, [\text{SEP}] \rangle$. Here, $w$ denotes words in the sentence, $SP_{fm}$ and $SP_{desc}$ are FMs and their descriptions. We obtain representations $\mathbf{H}_s \in \mathbb{R}^{N \times D}$ and $\mathbf{h}_d \in \mathbb{R}^{k \times D}$ by feeding $S$ and $Kn$ into LM1,

$$\mathbf{H}_s = \text{LM1}(S), \qquad \mathbf{h}_d = \text{LM1}(Kn) \quad (1)$$

where $N$ is the length of the input sentence, $k$ is the number of FMs contained in this sentence and $D$ is the hidden dimension. It should be pointed out that we treat the representation of [CLS] token of $Kn$ as $\mathbf{h}_d$. Subsequently, we select FM word representations from $\mathbf{H}_s$ as $\mathbf{h}_s \in \mathbb{R}^{k \times D}$, which are from plain sentences $S$. To integrate knowledge during pre-training, we coalesce the representations of FM words from both plain sentences $S$ and metric descriptions $Kn$. Specifically, we employ a learnable weight $\mathbf{W} \in \mathbb{R}^{2D \times D}$ to adaptively adjust the contributions of both $\mathbf{h}_d$ and $\mathbf{h}_s$ to obtain the fused representation $\mathbf{h}_c$, which is as follows.

$$\mathbf{h}_c = \mathbf{W}([\mathbf{h}_d, \mathbf{h}_s]), \qquad (2)$$

where $[\mathbf{h}_d, \mathbf{h}_s] \in \mathbb{R}^{k \times 2D}$ is the stacked matrix of $\mathbf{h}_d$ and $\mathbf{h}_s$. Next, we use $\mathbf{h}_c$ instead of the relevant part of the FM words $\mathbf{h}_s$ in $\mathbf{H}_s$ to form $\mathbf{H}'_s$, and feed $\mathbf{H}'_s$ into LM2 to continue pre-training.

Similarly, we employ the masked language model (MLM) objective to provide supervision signals for training both LM1 and LM2 simultaneously. Specifically, we randomly mask 15% of the tokens. When a token is masked, we substitute it with (1) the [MASK] token 80% of the time, (2) a randomly drawn token from the default glossary 10% of the time, (3) the unchanged token 10% of the time. We use the KePt dataset to further pre-train the BERT$_{\text{BASE}}$ model and save the parameters of this model for future utilization. For more implementation details, refer to Appendix C.

### 3.2 Financial Volatility Prediction (FVP)

We demonstrate the formulation of the financial volatility prediction task based on earnings calls. For each stock, there exist multiple earnings calls, which are held quarterly in cycles. In this study, we focus solely on the impact of one of earnings calls on subsequent stock price volatility. Each earnings call transcript $X_T = \langle x_1, \ldots, x_n \rangle$ consists of numerous sentences, where $n$ is the total number of sentences, detailed statistics are in Table 1. Given the input transcript $X_T$, we first employ a text encoder to map them into proper representation space. We combine LM1 and LM2 to act as

the text encoder TEXTENC. Specifically,

$$\mathbf{H}_T = \text{TEXTENC}(X_T), \qquad (3)$$

where $\mathbf{H}_T \in \mathbb{R}^{n \times d}$ denotes the representation of a transcript, and $d$ is the dimension of hidden layers.

Meanwhile, there also exists an adjusted closing price[8] for each stock on each trading day. Assuming that an earnings call is announced on the day $d$, we collect adjusted closing price series $\mathbf{P}_I = \langle p_{d-I}, \ldots, p_d \rangle$ for the time window $I$ prior to the announcement of earnings calls as historical price data. Subsequently, we calculate historical average log volatility series $\mathbf{V}_I = \langle v_1, \ldots, v_I \rangle$ based on $\mathbf{P}_I$ (refer to the Appendix A for details).

Given the transcript representation $\mathbf{H}_T$ and historical volatility series $\mathbf{V}_I$ for a specific stock, the objective of this task is to predict the average log volatility $v_{[d,d+\tau]}$ at the future time period $\tau$.

### 3.2.1 Information Enhancement (IE)

We adopt Transformer blocks to compose the information enhancement module TRANSIE. Concretely, we feed $\mathbf{H}_{ie}^0 = \mathbf{H}_T$ attached with sentence mask $\mathbf{M}_{sent}$ into $L$ Transformer blocks,

$$\mathbf{H}_{ie}^{L-1} = \text{TRANSIE}(\mathbf{H}_{ie}^0, \mathbf{M}_{sent}), \qquad (4)$$

where $\mathbf{M}_{sent}$ is to indicate whether the position is a real feature or a padding. For the $l$-th Transformer block, the computation is conducted as follows:

$$\mathbf{H}_{ie}^{\tilde{l}-1} = \text{SELFATT}_l(\mathbf{H}_{ie}^{l-1}, \mathbf{M}_{sent}), \qquad (5)$$

$$\mathbf{H}_{ie}^{\hat{l}-1} = \text{LN}_l^1(\mathbf{H}_{ie}^{l-1} + \mathbf{H}_{ie}^{\tilde{l}-1}), \qquad (6)$$

$$\mathbf{H}_{ie}^l = \text{LN}_l^2(\mathbf{H}_{ie}^{\hat{l}-1} + \text{FEEDFOWARD}(\mathbf{H}_{ie}^{\hat{l}-1})), \qquad (7)$$

where SELFATT denotes self-attention mechanism and LN denotes layer normalization (Vaswani et al., 2017). Then, we obtain the transcript representation $\mathbf{H}_{ie}^{L-1} \in \mathbb{R}^{n \times d}$, serving as the input to CTSP.

### 3.2.2 Conditional Time Series Prediction (CTSP)

To predict future volatility, we input historical volatility information $\mathbf{V}_I$ into the model, where $I$ is the total number of timestamps. To jointly process time series and text information $\mathbf{H}_{ie}^{L-1}$ after the IE module, we utilize a CTSP module based on Autoformer (Wu et al., 2021) to treat $\mathbf{H}_{ie}^{L-1}$ as a condition when making predictions. Here we define $\mathbf{H}_{cd}^0 = \mathbf{H}_{ie}^{L-1}$ to carry out the later operation.

---

[8]https://www.investopedia.com/terms/a/adjusted_closing_price.asp

As shown in Figure 1 (bottom(b)), we employ $N$ Autoformer encoders to model historical volatility series. Specifically, for the $i$-th Autoformer encoder $\text{AUTOENC}_i(\cdot)$, the calculation is as follows:

$$\mathbf{V}_{En}^0 = \mathbf{V}_I, \mathbf{V}_{En}^i = \text{AUTOENC}_i(\mathbf{V}_{En}^{i-1}). \quad (8)$$

The output feature $\mathbf{V}_{En}^i \in \mathbb{R}^{I \times d}$ will be provided to Autoformer decoder $\text{AUTODEC}(\cdot)$ and conditional attention module $\text{CONDATT}(\cdot)$ for prediction. We encapsulate $\text{AUTODEC}(\cdot)$ and $\text{CONDATT}(\cdot)$ as a conditional decoder, and we employ $M$ conditional decoders. Following (Wu et al., 2021), we also employ a series decomposition module to decompose the time series into trend and seasonal items ($\mathbf{V}_t, \mathbf{V}_s \in \mathbb{R}^{I \times d}$) (refer to Appendix D for details).

For $j$-th Autoformer decoder $\text{AUTODEC}_j(\cdot)$, we use $\mathbf{H}_{cd}^{j-1}$ to fill in the part to be predicted in the seasonal item $\mathbf{V}_{s,De}^{j-1}$ to form the conditional seasonal item $\mathbf{V}_{s,C}^{j-1}$. Then, we feed $\mathbf{V}_{s,C}^{j-1}$ along with $\mathbf{V}_{En}^i$ into $\text{AUTODEC}_j(\cdot)$.

$$\mathbf{V}_{s,C}^{j-1} = [\mathbf{V}_{s,De}^{j-1}; \mathbf{H}_{cd}^{j-1}], \qquad (9)$$

$$\mathbf{V}_{t,De}^j, \mathbf{V}_{s,De}^j = \text{AUTODEC}_j(\mathbf{V}_{s,C}^{j-1}, \mathbf{V}_{En}^i), \quad (10)$$

$$\mathbf{V}_t^j = \mathbf{V}_t^{j-1} + \mathbf{V}_{t,De}^j \qquad (11)$$

where $\mathbf{V}_{s,De}^0 = \mathbf{V}_s$, $\mathbf{V}_t^0 = \mathbf{V}_t$, and $[;]$ is the concatenation operation. To further capture the interaction between text and historical volatility information, we apply the $\text{CONDATT}_j(\cdot)$ module to fuse $\mathbf{H}_{cd}^{j-1}$ and $\mathbf{V}_{En}^i$ beside $\text{AUTODEC}_j(\cdot)$.

$$\mathbf{H}_{cd}^j = \text{CONDATT}_j(\mathbf{H}_{cd}^{j-1}, \mathbf{V}_{En}^i, \mathbf{V}_{En}^i). \qquad (12)$$

Specifically,

$$\begin{aligned} \mathbf{Q} &= Linear_q(\mathbf{H}_{cd}^{j-1}), \\ \mathbf{K}, \mathbf{H}_V &= Linear_k(\mathbf{V}_{En}^i), Linear_v(\mathbf{V}_{En}^i), \\ \mathbf{H}_{cd}^j &= \mathbf{H}_{cd}^{j-1} + \|_{u=1}^U softmax(\frac{\mathbf{Q}\mathbf{K}^\top}{\sqrt{d}})\mathbf{H}_V, \end{aligned} \quad (13)$$

where $\mathbf{H}_{cd}^j$ is the output of $\text{CONDATT}_j(\cdot)$. Then, we also use $\mathbf{H}_{cd}^j$ to fill in the part to be predicted in the seasonal item $\mathbf{V}_{s,De}^j$ of $\text{AUTODEC}_j(\cdot)$.

$$\mathbf{V}_{s,C}^j = [\mathbf{V}_{s,De}^j; \mathbf{H}_{cd}^j], \qquad (14)$$

where $\mathbf{V}_{s,C}^j$ is the conditional seasonal item for the next Autoformer decoder.

Table 2: The overall performance. The results with ♮, ♯ and ♭ are retrieved from (Yang et al., 2022), (Li et al., 2020) and (Sawhney et al., 2020c) respectively, and the remainder except for KeFVP are from (Sawhney et al., 2020b). KeFVP is the average result across 10 runs. KeFVP(Best) reports the best result in these 10 runs. $MSE_{3\sim30}$ are MSE scores of different time periods, and $\overline{MSE}$ is the average over above. The form of *A(B)* denotes mean (for A) and standard deviation (for B). The best results are in bold, and the second-best restuls are underlined.

| Model | EC | | | | | MAEC-15 | | | | | MAEC-16 | | | | |
|---|---|---|---|---|---|---|---|---|---|---|---|---|---|---|---|
| | $\overline{MSE}$ | $MSE_3$ | $MSE_7$ | $MSE_{15}$ | $MSE_{30}$ | $\overline{MSE}$ | $MSE_3$ | $MSE_7$ | $MSE_{15}$ | $MSE_{30}$ | $\overline{MSE}$ | $MSE_3$ | $MSE_7$ | $MSE_{15}$ | $MSE_{30}$ |
| Vpast | 1.12 | 2.99 | 0.83 | 0.42 | 0.23 | - | - | - | - | - | - | - | - | - | - |
| Price LSTM | 0.75 | 1.97 | 0.46 | 0.32 | 0.24 | - | - | - | - | - | - | - | - | - | - |
| BiLSTM + ATT | 0.74 | 1.98 | 0.44 | 0.30 | 0.23 | 0.696 | 1.599♯ | 0.560♯ | 0.339♯ | 0.284♯ | 0.691 | 1.544♯ | 0.571♯ | 0.362♯ | 0.288♯ |
| HAN(Glove) | 0.60 | 1.43 | 0.46 | 0.31 | 0.20 | - | - | - | - | - | - | - | - | - | - |
| MDRM(Audio) | 0.60 | 1.41 | 0.44 | 0.32 | 0.22 | - | - | - | - | - | - | - | - | - | - |
| MDRM(Text+Audio) | 0.58 | 1.37 | 0.42 | 0.30 | 0.22 | 0.630 | 1.425♯ | 0.488♯ | 0.320♯ | 0.285♯ | 0.618 | 1.426♯ | 0.476♯ | 0.311♯ | 0.259♯ |
| HTML(Text) | 0.46 | 1.18 | 0.37 | **0.15** | 0.13 | 0.514 | 1.199♯ | 0.440♯ | 0.231♯ | 0.187♯ | 0.579 | 1.287♯ | 0.479♯ | **0.300**♯ | 0.249♯ |
| HTML(Text+Audio) | 0.40 | 0.85 | 0.35 | 0.25 | 0.16 | 0.487 | 1.065♯ | 0.416♯ | 0.272♯ | 0.196♯ | 0.556 | 1.160♯ | 0.515♯ | 0.314♯ | 0.236♯ |
| VolTAGE | 0.31 | 0.63 | **0.29** | 0.17 | 0.14 | - | - | - | - | - | - | - | - | - | - |
| **KeFVP** | **0.300** | **0.610** (3.31e-2) | **0.291** (1.33e-2) | 0.183 (0.89e-2) | **0.114** (0.63e-2) | 0.204 | 0.418 (1.23e-2) | 0.187 (0.27e-2) | 0.122 (0.32e-2) | 0.087 (0.17e-2) | 0.318 | 0.445 (6.36e-2) | 0.279 (4.42e-2) | 0.303 (3.65e-2) | 0.177 (3.33e-2) |
| SVM(TF-IDF)♭ | 0.70 | 1.70 | 0.50 | 0.34 | 0.25 | - | - | - | - | - | - | - | - | - | - |
| bc-LSTM♭ | 0.59 | 1.42 | 0.44 | 0.30 | 0.22 | - | - | - | - | - | - | - | - | - | - |
| Multi-Fusion CNN♭ | 0.41 | 0.73 | 0.35 | 0.29 | 0.28 | - | - | - | - | - | - | - | - | - | - |
| NumHTML(Text+Audio)♮ | 0.31 | - | - | - | - | - | - | - | - | - | - | - | - | - | - |
| Ensemble(Text+Audio)♭ | 0.302 | 0.601 | 0.308 | 0.181 | 0.119 | - | - | - | - | - | - | - | - | - | - |
| **KeFVP(Best)** | **0.276** | **0.565** | **0.265** | 0.171 | **0.101** | 0.198 | 0.407 | 0.182 | 0.117 | 0.084 | 0.245 | 0.347 | 0.194 | 0.223 | 0.126 |

### 3.3 Model Training and Inference

In the end, we add up the final obtained $\mathbf{V}_{s,C}^{M-1}$ and $\mathbf{V}_t^{M-1}$, and employ a fully connected layer as the prediction layer. Specifically,

$$\hat{y_m} = \mathbf{W}_p(\mathbf{V}_{s,C}^{M-1} + \mathbf{V}_t^{M-1}) + \mathbf{b}_p, \quad (15)$$

where $\mathbf{W}_p \in \mathbb{R}^{d \times 1}$ and $\mathbf{b}_p \in \mathbb{R}^{1 \times 1}$ are the weight matrix and bias, respectively. The objective is:

$$\mathcal{L} = \sum_D \Big( (\hat{y_m} - y_m)^2 \Big), \quad (16)$$

where $\hat{y_m}$ is the predicted volatility, and $y_m$ is the ground truth.

## 4 Experiments

### 4.1 Datasets

Following previous works, our experiments are conducted on EC (Qin and Yang, 2019), MAEC-15, and MAEC-16 (Li et al., 2020) datasets (refer to Appendix B for details). The statistics are displayed in Table 1. We partition each dataset into training, validation, and testing sets in a 7:1:2 ratio, consistent with prior works. We experiment on settings of n $\in \{3, 7, 15, 30\}$ days to explore short- to medium- and long-term performance. For implementation details, refer to Appendix C.

### 4.2 Baselines

We group baselines according to the information (price, text, and audio) they used for prediction.

**Price:** Vpast (Qin and Yang, 2019), Price LSTM (Kim and Won, 2018) and BiLSTM + ATT (Siami-Namini et al., 2019);

**Price and text:** HAN (Hu et al., 2021), HTML(Text) (Yang et al., 2020), SVM(TF-IDF) (Tsai and Wang, 2014; Ding et al., 2014);

**Price, text, and audio:** bc-LSTM (Poria et al., 2017), Multi-Fusion CNN (Sebastian and Pierucci, 2019), MDRM(Text+Audio) (Qin and Yang, 2019), Ensemble(Text+Audio) (Sawhney et al., 2020c), NumHTML(Text+Audio) (Yang et al., 2022), HTML(Text+Audio) (Yang et al., 2020), VolTAGE (Sawhney et al., 2020b).

### 4.3 Main Results

As shown in Table 2, we report the main results compared with baselines. Following (Sawhney et al., 2020b; Yang et al., 2020; Qin and Yang, 2019), we chose MSE as the comparative metric (refer to the Appendix C for details). For a fair comparison, we report the average results of 10 runs and the best results of those 10 runs, as some baselines (Sawhney et al., 2020b) report average results, while others (Yang et al., 2022; Li et al., 2020; Sawhney et al., 2020c) do not. We make predictions for 3, 7, 15, and 30-day periods following previous works. Although on the EC dataset, for the $MSE_{15}$ results, our method did not perform as well as expected, KeFVP outperformed all baselines in terms of both average and best results for other time periods. For day 7 of the EC dataset and day 15 of the MAEC-16 dataset, KeFVP is basically on par with VolTAGE in terms of average results but exceeded its performance in terms of best results. In addition, it is worth noting that KeFVP outperforms all models that incorporate both text and audio information when only text is uti-

Table 3: Results of ablation study on the EC dataset. **KeFVP** denotes our overall method; KeFVP(w/o CTSP & IE) denotes removing CTSP and IE, and only text remains; KeFVP(w/o Text) denotes removing text only historical price remains; KeFVP(w/o IE) denotes removing information enhancement (IE); KeFVP(w/o X) denotes using text embedding from raw $\text{BERT}_{\text{BASE}}$ ($X = KePt$) and adaptively pre-trained PLMs without knowledge injection ($X = Knowledge$).

| Pattern | Model | $\overline{MSE}$ | $\text{MSE}_3$ | $\text{MSE}_7$ | $\text{MSE}_{15}$ | $\text{MSE}_{30}$ |
|---|---|---|---|---|---|---|
| FVP | KeFVP(w/o CTSP & IE) | 0.392 | 0.743(2.94e-2) | 0.380(1.72e-2) | 0.258(1.43e-2) | 0.185(1.17e-2) |
| | KeFVP(w/o Text) | 0.331 | 0.647(2.55e-2) | 0.328(1.52e-2) | 0.210(1.32e-2) | 0.138(1.35e-2) |
| | KeFVP(w/o IE) | 0.319 | 0.642(4.80e-2) | 0.308(2.08e-2) | 0.198(1.60e-2) | 0.129(1.38e-2) |
| KePt | KeFVP(w/o KePt) | 0.318 | 0.644(2.52e-2) | 0.311(0.66e-2) | 0.196(1.23e-2) | 0.120(0.60e-2) |
| | KeFVP(w/o Knowledge) | 0.319 | 0.650(3.11e-2) | 0.310(1.19e-2) | 0.195(1.33e-2) | 0.120(0.69e-2) |
| Overall | **KeFVP** | **0.300** | **0.610(3.31e-2)** | **0.291(1.33e-2)** | **0.183(0.89e-2)** | **0.114(0.63e-2)** |

lized, which proves the superiority of our method.

## 4.4 Ablation Study

**For KePt.** As shown in Table 3, we conduct ablation studies to substitute KePt with text embedding counterparts: KeFVP(w/o KePt) and KeFVP(w/o Knowledge). Relative to KeFVP, the effects of both KeFVP(w/o Knowledge) and KeFVP(w/o KePt) decrease, which indicates that neither the adaptive pre-training without knowledge nor the direct use of the raw published $\text{BERT}_{\text{BASE}}$ is as effective as that using KePt.

**For FVP.** To illustrate the effects of various components, ablation experiments are carried out on the EC dataset as shown in Table 3. KeFVP(w/o IE) drops significantly when we remove IE. It can be concluded that IE plays a significant role in enhancing performance. For KeFVP(w/o Text), we exclude the effect of text information, meaning that only time series information is used. As can be observed, KeFVP(w/o Text) becomes significantly worse compared to KeFVP. Therefore, only historical information is insufficient, and combining it with text information will help prediction. This also illuminates the capability of CTSP to effectively amalgamate text and time series information, yielding robust predictions. Furthermore, we investigate the prediction performance relying solely on text information. Concretely, we substitute the CTSP and IE modules with a fully connected layer and exclusively use text information as input for prediction (i.e. KeFVP(w/o CTSP & IE)). It can be observed that KeFVP(w/o CTSP & IE) decreases a lot compared to KeFVP. Also, KeFVP(w/o CTSP & IE) drops substantially in contrast to KeFVP(w/o IE). This demonstrates the significance of CTSP.

Table 4: Comparison with financial PLMs on the EC dataset. FVP(PLMs) denotes using text embedding from the corresponding PLMs. Corpus Size of FVP(KePt-BERT) and FVP(FinBERT) are counted by the number of sentences in each corpus, while FVP(FLANG-BERT) is counted by the number of documents in each corpus.

| Model | Corpus Size | $\overline{MSE}$ | $\text{MSE}_3$ | $\text{MSE}_7$ | $\text{MSE}_{15}$ | $\text{MSE}_{30}$ |
|---|---|---|---|---|---|---|
| FVP($\text{BERT}_{\text{BASE}}$) | - | 0.318 | 0.644(2.52e-2) | 0.311(0.66e-2) | 0.196(1.23e-2) | 0.120(0.60e-2) |
| FVP(FinBERT) | 406,019 | 0.318 | 0.631(2.22e-2) | 0.322(0.72e-2) | 0.195(1.19e-2) | 0.122(0.74e-2) |
| FVP(FLANG-BERT) | 696,001 | 0.313 | 0.644(5.75e-2) | 0.293(1.86e-2) | 0.190(0.78e-2) | 0.124(0.59e-2) |
| **FVP(KePt-BERT)** | **8,732** | **0.300** | **0.610(3.31e-2)** | **0.291(1.33e-2)** | **0.183(0.89e-2)** | **0.114(0.63e-2)** |

## 4.5 Financial PLMs

In this section, we compare the performance of our KePt-BERT with two popular BERT-based financial PLMs (FinBERT (Araci, 2019) and FLANG-BERT (Shah et al., 2022)) on the volatility prediction task. The results are presented in Table 4.

FVP(FinBERT) is worse than FVP(KePt-BERT) in this task. We denote KeFVP as FVP(KePt-BERT) to clearly indicate the PLMs it utilizes. Note that FVP(FinBERT) is built on FinBERT (Araci, 2019), further training $\text{BERT}_{\text{BASE}}$ (Devlin et al., 2019) on large financial corpora (consists of TRC2-financial, Financial PhraseBank, and FiQA dataset) (containing approximately 406,019 sentences in total) but neglecting FM knowledge, whereas FVP(KePt-BERT) using KePt is based on a much smaller corpus (KePt dataset, containing 8,732 sentences) with FM knowledge. This observation underscores the potential of enhancing the efficacy of volatility prediction through the utilization of KePt to inject knowledge.

Moreover, we conducted experiments involving FLANG-BERT, designated as FVP (FLANG-BERT). Notably, it's important to acknowledge that the dataset used for training KePt-BERT is also notably smaller than that of FLANG-BERT. The specific corpus sizes are provided in detail within the Table 4. It is apparent that in this task, both FinBERT and FLANG-BERT exhibit less favorable

Table 5: Comparison with non-text baselines.

| Model | $\overline{MSE}$ | $MSE_3$ | $MSE_7$ | $MSE_{15}$ | $MSE_{30}$ |
|---|---|---|---|---|---|
| Linear Regression | 0.622 | 0.995(3.20e-2) | 0.595(5.83e-2) | 0.495(3.90e-2) | 0.402(3.80e-2) |
| GARCH | 1.756 | 2.084(1.88) | 1.729(1.78) | 1.657(1.85) | 1.555(1.83) |
| ARIMA | 0.611 | 0.944(6.94e-1) | 0.596(6.66e-1) | 0.497(7.10e-1) | 0.407(5.78e-2) |
| KeFVP(w/o Text) | 0.331 | 0.647(2.55e-2) | 0.328(1.52e-2) | 0.210(1.32e-2) | 0.138(1.35e-2) |
| **KeFVP** | **0.300** | **0.610(3.31e-2)** | **0.291(1.33e-2)** | **0.183(0.89e-2)** | **0.114(0.63e-2)** |

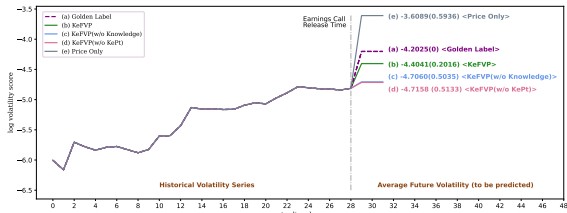

Figure 2: Impact of text embeddings. The form of P(Q) denotes predicted results (for P) and margins (for Q).

performance compared to KePt-BERT.

## 4.6 Non-text Baselines

Furthermore, we extend our experiments on the EC dataset to include non-text baselines, namely the classical Linear Regression, GARCH, and ARIMA models. These additional baselines are presented in the Table 5. Due to data availability constraints and the need to ensure comparability, we have maintained the use of historical price information as non-textual data, consistent with previous works. Notably, we have refrained from incorporating textual data from earnings calls in this context.

Regarding the interpretability of transformer-based models, regression models offer intuitive explanations due to their fewer parameters. However, their limited fitting capacity restricts their effectiveness. Transformer-based models exhibit robust fitting capabilities, albeit with a larger parameter count. These models can be explained visually through methods like attention visualization to some extent, which aids in the intuitive understanding of a substantial number of parameters and thereby contributes to result comprehension.

Simultaneously, we introduced KeFVP (w/o Text) as a point of comparison. This outcome represents the results derived solely from the Autoformer model's treatment of time series data. In comparison to KeFVP, it is evident that the incorporation of textual information yields a significant enhancement in predictive performance compared with dealing with time series data in isolation.

## 4.7 Case Study

As shown in Figure 2, we visually exhibit the influence of different text embeddings for volatility predictions. We select the case related to Fidelity National Information Services (FIS Inc), an American multinational corporation that offers financial products and services, from the EC dataset. All analyses are based on the 3-day prediction. Figure 2(a) is the golden label of this case. Day 28 (the light gray vertical dashed line in the chart) marks the release date of the earnings call, with the subsequent period constituting the target of our

prediction, and the preceding period representing the historical volatility series. Figure 2(b-e) demonstrate the predicted results of KeFVP, KeFVP(w/o Knowledge), KeFVP(w/o KePt), and Price Only (relying solely on the historical price), respectively. The prediction of KeFVP (in Figure 2(b)) is the closest to the golden label. This proves that KePt is the most efficacious of all counterparts. The margin between the predicted result ($-3.6089$) and golden label ($-4.2025$) in Figure 2(e) is 0.5936, which is substantially greater than KeFVP (the margin is 0.2016), and also larger than the other counterparts (KeFVP(w/o Knowledge) (0.5035) and KeFVP(w/o KePt) (0.5133)). This indicates that combining information from time series and text data can help predictions. Still, more importantly, the different levels of understanding of text data (different text embeddings) play a great role in prediction.

## 4.8 Evaluation on KePt

Financial sentiment analysis serves as a cornerstone in the realm of financial text mining. This task is dedicated to scrutinizing the emotional dynamics within the financial market, and its efficacy is anchored in a deep understanding of financial texts. In this section, we apply our KePt method to financial sentiment analysis as a further testament to its effectiveness. We employ three standard datasets for this purpose: FiQA-headline, FiQA-post, and PhraseBank. Comprehensive statistics along with experimental settings are elaborated in Appendix E. For the FiQA-headline and FiQA-post datasets, our performance metrics are Mean Squared Error (MSE) and R Square ($R^2$). For the PhraseBank dataset, we gauge performance using Accuracy (Acc) and Macro-f1 (F1) scores. The ensuing results are encapsulated in Table 6.

We compare KePt with two baselines: KePt(w/o Knowledge) and BERT$_{BASE}$. In all datasets, KePt outperforms, further attesting to its potency. When comparing KePt and KePt(w/o Knowledge) on the

Table 6: The results of KePt on financial sentiment analysis. KePt(w/o Knowledge) denotes adaptively pre-training on the KePt dataset without knowledge injection.

| Model | FiQA-headline | | FiQA-post | | PhraseBank | |
|---|---|---|---|---|---|---|
| | MSE ↓ | $R^2$ ↑ | MSE ↓ | $R^2$ ↑ | Acc(%) ↑ | F1(%) ↑ |
| BERT$_{BASE}$ | 0.115 | 0.297 | 0.101 | 0.342 | 80.54 | 82.97 |
| KePt(w/o Knowledge) | 0.100 | 0.297 | 0.084 | 0.438 | 78.40 | 82.56 |
| KePt | **0.080** | **0.435** | **0.077** | **0.488** | **82.73** | **83.80** |

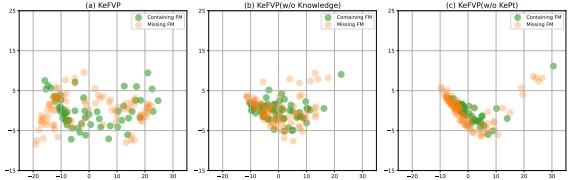

Figure 3: Visualization of sentence embeddings. Containing FM denotes sentences with FMs while Missing FM refers to those without them.

FiQA-headline and FiQA-post, the improvement for MSE and $R^2$ are 0.020 and 0.138 on FiQA-headline, and the improvement for these metrics are 0.007 and 0.05 on FiQA-post. These improvements align with the *FM Ratio* (17.89% for FiQA-headline and 9.19% for FiQA-post) presented in Table 10 in Appendix E.1. KePt exerts a more pronounced effect on datasets with a higher *FM Ratio*, indicating the positive influence of incorporating FM knowledge on text understanding.

### 4.9 Sentence Embedding Visualization

To further elucidate the effect of KePt, this section provides visual demonstrations of sentence embeddings from three ablation models: (a) KeFVP, (b) KeFVP(w/o Knowledge), and (c) KeFVP(w/o KePt). As shown in Figure 3, this case is from the EC dataset, the earnings call transcript issued by Fidelity National Information Services (FIS Inc) on February 7, 2017. The total sentence number of this transcript is 147, and there are 58 sentences containing FM and 89 sentences not containing. We apply PCA (Principal Component Analysis) to reduce the dimensionality of sentence embeddings to represent them in a two-dimensional space. Comparing Figure 3(a) with (b) and (c), we can observe that Figure 3(a) has a more dispersed distribution whether for sentences containing FM or sentences missing FM. So it can be inferred that injecting FM knowledge can not only improve the representation of sentences containing FM but also improve the representation ability of sentences missing FM.

Table 7: Comparison of time series models on the EC dataset. KeFVP(Y) denotes the KeFVP equipped with different time series models (CondTF = Conditional Transformer, CondLSTM = Conditional LSTM).

| Model | $\overline{MSE}$ | MSE$_3$ | MSE$_7$ | MSE$_{15}$ | MSE$_{30}$ |
|---|---|---|---|---|---|
| KeFVP(CondTF) | 0.388 | 0.731(4.34e-2) | 0.368(1.06e-2) | 0.254(1.04e-2) | 0.198(0.87e-2) |
| KeFVP(CondLSTM) | 0.327 | 0.681(4.53e-2) | 0.320(2.86e-2) | 0.195(1.44e-2) | **0.113**(0.90e-2) |
| **KeFVP(CTSP)** | **0.300** | **0.610(3.31e-2)** | **0.291 (1.33e-2)** | **0.183 (0.89e-2)** | 0.114 (0.63e-2) |

### 4.10 Impact of Different Time Series Models

To highlight the effect of CTSP, we conduct experiments employing different time series models instead of Autoformer. We replace Autoformer with Transformer by removing the series decomposition and replacing auto-correlation (Wu et al., 2021) with self-attention, and the remaining operations are in line with the rest of CTSP. We refer to this model as the conditional Transformer (CondTF). As demonstrated in Table 7, there is a noticeable performance gap between KeFVP(CondTF) and KeFVP(CTSP), which illustrates merging price and text information under Autoformer is more effective. In addition, we borrow the Conditional LSTM (Sawhney et al., 2020b) as another counterpart, which we called KeFVP(CondLSTM). Overall, KeFVP(CTSP) is better than KeFVP(CondLSTM). For $MSE_{30}$, KeFVP(CondLSTM) is comparable to KeFVP(CTSP), while the standard deviation of KeFVP(CondLSTM) is much larger than KeFVP(CTSP), which indicates that the stability of KeFVP(CTSP) is superior to KeFVP(CondLSTM). We also add further case studies in Appendix F to explore the impact of textual information on time series predictions.

### 5 Conclusion and Future Work

In this work, we present KeFVP and systematically illustrate the FM knowledge introduction into predictions. The KePt method is proposed and the KePt dataset is concurrently constructed to serve it. The FVP equipped with IE and CTSP modules to integrate text and price information drives results to SOTA. Further, this method is also informative for research based on other financial text (e.g. news, tweets, etc.). In the future, we will explore ways to incorporate other financial knowledge (e.g., financial analysis formulas) into financial applications.

### Limitations

While our method underscores the significance of incorporating FM knowledge to enhance volatility

predictions and financial sentiment analysis, it is not without limitations that should be addressed.

Primarily, the scope of our training data has constrained the extent of performance improvement. Despite the advancements, the limited size of our dataset has inevitably affected our model's capacity to perform optimally. Some future work will focus on augmenting the volume of training data to further enhance model performance.

Secondly, our method is fundamentally built on encoder-based pre-trained models (BERT). However, the applicability of our approach to other model architectures like decoder-based (e.g., GPT series) and encoder-decoder-based (e.g., T5 series) models remains unexplored. We postulate that the introduction of FM knowledge may still be beneficial for these models, and hence, exploring this is a promising direction for future research.

## Acknowledgements

This work is funded in part by the National Natural Science Foundation of China Project (No.U1936213), the Shanghai Science and Technology Development Fund (No.22dz1200704), and CNKLSTISS.

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

Table 8: Statistics on dataset KePt. The overall number of sentences in KePt is under *#Total Sent.*, and the *Ave. Length* denotes the average length of sentences. The overall number of FMs in KePt and the average number of FMs in each sentence are under *#Total FM.* and *# Ave. FM.*, respectively.

| Dataset | #Total Sent. | Ave. Length | #Total FM. | #Ave. FM. |
|---------|--------------|-------------|------------|-----------|
| KePt | 8732 | 156.19 | 11442 | 1.31 |

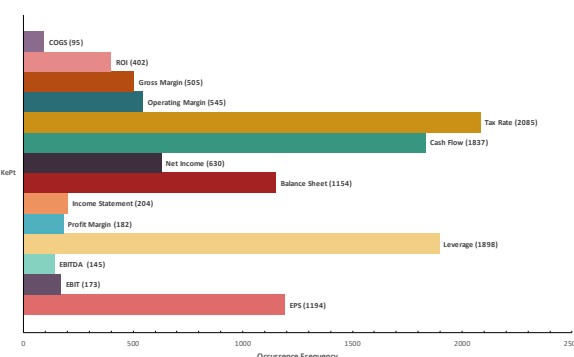

Figure 4: Occurrence frequency of main FMs on the KePt dataset.

## A Volatility Calculation

Following (Sawhney et al., 2020b), for a specific stock, its adjusted closing price on the trading day $j$ is $p_j$, and the average log volatility over trading day $d_{st}$ to trading day $d_{st} + \delta$ is calculated as follows:

$$v_{[d_{st}, d_{st}+\delta]} = \ln\left(\sqrt{\frac{\sum_{j=d_{st}}^{d_{st}+\delta}(r_j - \bar{r})^2}{\delta}}\right), \quad (17)$$

where the return of trading day $j$ is defined as $r_j = \frac{p_j}{p_{j-1}} - 1$, and $\bar{r}$ is the mean of return over trading day $d_{st}$ to trading day $d_{st} + \delta$.

For the historical price data $\mathbf{P}_I = \langle p_{d-I}, \ldots, p_d \rangle$, we follow previous works (Qin and Yang, 2019; Sawhney et al., 2020c; Yang et al., 2020; Sawhney et al., 2020b) to calculate the historical average log volatility $\mathbf{V}_I = \langle v_{[d-I,d-I+1]}, \ldots, v_{[d-I,d]}] \rangle$ for the time window $I$ according to Equation (17), where $d_{st} = d - I$ and $\delta \in \{1, 2, \ldots, I\}$. For simplicity, we abbreviate $\mathbf{V}_I$ as $\mathbf{V}_I = \langle v_1, \ldots, v_I \rangle$ taking the values of $\delta$ as subscripts.

## B Dataset Analysis

**EC** Following previous works, our experiments are conducted on the EC dataset, a publicly available earnings call dataset released by (Qin and Yang, 2019). The dataset contains 559 earnings call transcripts for 277 S&P 500 companies. Each

transcript is divided into a series of sentences, and the detailed statistics are displayed in Table 1. In addition, intuitively, we also illustrate the statistics of major financial metrics (FMs) for this dataset in Table 9. The stock prices (time series data) are extracted from Yahoo Finance [9] in the time frame from 1 January'17 to 31 December'17.

**MAEC-15, MAEC-16** Following the previous work (Li et al., 2020), we also conduct experiments on the datasets published in their work, which we refer to as MAEC-15 and MAEC-16. As they do not publish processed historical price data, we grabbed the corresponding historical prices for the period when the earnings call was issued via Yahoo Finance (we took a time window of 30 days before and after). We then use the volatility formula (17) to calculate the volatility of these prices. We will also release the processed price dataset in our source code.

We also provide detailed statistics of the KePt dataset and main FMs for both the three earnings call datasets and KePt dataset. The detailed statistic of KePt dataset is displayed in Table 8. The occurrence frequency and detailed description of each main FM for both earnings call datasets and the KePt dataset are listed in Table 9. Also, for visual illustration, we provide distribution maps of the main FMs over these datasets (Figure 6 and Figure 4). It can be found that the frequency distribution of FMs is basically the same in all datasets.

## C Implementation Details

The KePt is based on the pre-trained $\text{BERT}_{\text{BASE}}$ (Devlin et al., 2019), the BERT containing 12 hidden layers, and 768 hidden dimensions for each layer. The number of the epoch is 60 for KePt. We use the AdamW optimizer while training with the learning rate initialized by 2e-5. For FVP, we still employ the Adam optimizer (Kingma and Ba, 2015) and initialize the learning rate to 2e-4, weight decay to 0.05, and the number of training epochs is 200. We conduct experiments on two NVIDIA GeForce GTX 1080Ti, and our codes are implemented based on Pytorch. The average computation time on the GPU (1080Ti) is 5 hours for KePt and 15 minutes for FVP. Our source code for both KePt and FVP will be released later.

Following (Sawhney et al., 2020b; Yang et al., 2020; Qin and Yang, 2019), we treat the volatility

---

[9] https://finance.yahoo.com

Table 9: Statistics of the main FMs for three public datasets and our KePt dataset. The *Description* is the explanation extracted from Wikidata for a specific FM, and the *Frequency* denotes the number of times the FMs appear in the dataset. Here we only list the FMs that appear more frequently.

| FMs | Descriptions | Frequency | | | |
|---|---|---|---|---|---|
| | | EC | MAEC-15 | MAEC-16 | KePt |
| EPS | Earnings per share, value of earnings per outstanding share of common stock for a company | 978 | 337 | 628 | 1194 |
| EBIT | Earnings before interest and taxes, measure of a firm's profit | 121 | 25 | 56 | 173 |
| EBITDA | Accounting measure: net earnings, before interest expenses, taxes, depreciation, and amortization are subtracted | 614 | 377 | 665 | 145 |
| PE | Price–earnings ratio, the ratio of a company's share price to the company's earnings per share | 16 | - | 8 | 238 |
| ROI | Return on investment, ratio between the net profit and cost of investment resulting from an investment of some resources | 34 | 24 | 26 | 402 |
| COGS | Cost of goods sold, carrying value of goods sold during a particular period | 31 | 6 | 7 | 95 |
| ROA | Return on assets, ratio to express the profitability of a company's assets in generating income | 11 | 1 | 9 | 13 |
| Leverage | The use of borrowed funds rather than fresh equity in the purchase of an asset | 552 | 414 | 674 | 1898 |
| Gearing | Leverage, the use of borrowed funds rather than fresh equity in the purchase of an asset | 4 | 4 | 13 | 50 |
| Profit Margin | Profit margin is the ratio between turnover and profit, in other words, what percentage of turnover remains as profit for the company | 50 | 39 | 62 | 182 |
| Income Statement | Financial statement of a company: shows the company's revenues and expenses during a particular period | 40 | 26 | 59 | 204 |
| Balance Sheet | Accounting financial summary | 440 | 283 | 606 | 1154 |
| Net Income | Measure of the profitability of a business venture | 234 | 210 | 463 | 630 |
| Cash Flow | Movement of money into or out of a business, PROJect, or financial product | 727 | 431 | 774 | 1837 |
| Tax Rate | Ratio (usually expressed as a percentage) at which a business or person is taxed | 597 | 195 | 354 | 2085 |
| Operating Margin | Relating operating profits to net sales | 527 | 157 | 207 | 545 |
| Gross Margin | Relating gross profits to net sales | 473 | 323 | 624 | 505 |

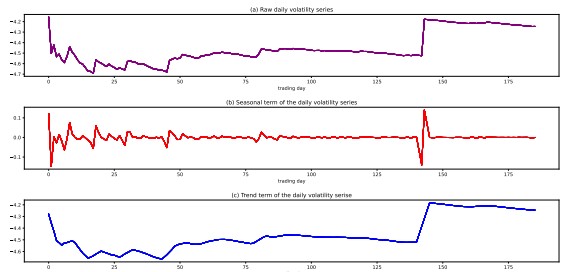

Figure 5: Time series decomposition of Amazon.com Inc daily volatility series from 2017/04/01 to 2017/12/31. Subfigure (a) is the volatility series without decomposition, and (b) and (c) are the decomposed seasonal and trend series, respectively.

prediction as a regression task, and we also chose the Mean Square Error (MSE) as the comparative metric as in previous works. The MSE is computed as follows:

$$MSE = \frac{\sum_{i=1}^{n}(\hat{y}_i - y_i)^2}{n} \quad (18)$$

where $\hat{y}_i$ is the predicted volatility, $y_i$ is the ground truth.

## D  Time Series Decomposition

The decomposition process is:

$$\begin{aligned} \mathbf{V}_t &= \text{AvgPool}(\text{Padding}(\mathbf{V}_I)), \\ \mathbf{V}_s &= \mathbf{V}_I - \mathbf{V}_t, \end{aligned} \quad (19)$$

where $\mathbf{V}_t, \mathbf{V}_s \in \mathbb{R}^{I \times d}$ denote trend and seasonal items, respectively. $\text{AvgPool}(\cdot)$ denotes the moving average pooling, and $\text{Padding}(\cdot)$ is to keep the time series length constant.

As shown in Figure 5, such time series decomposition makes seasonality more apparent and facilitates the merging with text information. Intuitively,

Table 10: Statistics on the three datasets for financial sentiment analysis. *#Sent.* denotes the total number of sentences in each dataset. The overall number of FMs in each dataset is under *# FM*, and *FM Ratio* is the ratio of the previous two. In addition, *FM Sent.* denotes the number of sentences containing FMs, and *FM / FM Sent.* indicates the average number of FMs in each *FM Sent.*.

| Dataset | | # Sent. | # FM | FM Ratio (%) | #FM Sent. | FM / FM Sent. |
|---|---|---|---|---|---|---|
| FiQA-headline | Train | 305 | 61 | 20.00 | 59 | 1.03 |
| | Test | 87 | 7 | 8.05 | 7 | 1 |
| | Dev. | 44 | 10 | 22.73 | 9 | 1.11 |
| | Overall | 436 | 78 | 17.89 | 75 | 1.04 |
| FiQA-post | Train | 473 | 50 | 10.57 | 45 | 1.11 |
| | Test | 135 | 15 | 11.11 | 13 | 1.15 |
| | Dev. | 67 | 4 | 5.97 | 4 | 1 |
| | Overall | 675 | 69 | 9.19 | 62 | 1.11 |
| PhraseBank | Train | 3392 | 784 | 23.11 | 674 | 1.16 |
| | Test | 969 | 205 | 21.16 | 176 | 1.16 |
| | Dev. | 484 | 112 | 23.14 | 92 | 1.22 |
| | Overall | 4845 | 1101 | 22.72 | 942 | 1.17 |

seasonal series excluding trend factors are more reflective of volatility, and their integration with text information will likely amplify this advantage. For the sake of such reasons, we use $\mathbf{H}_{cd}^{j-1}$ to fill in the part to be predicted in the seasonal item $\mathbf{V}_{s,De}^{j-1}$ to form the conditional seasonal item $\mathbf{V}_{s,C}^{j-1}$ for the $j$-th Autoformer decoder $\text{AutoDec}(\cdot)$.

## E  Financial Sentiment Analysis

### E.1  Dataset Statistics

As shown in Table 10, we performed statistics on the three financial sentiment analysis datasets (FiQA-headline, FiQA-post [10], and PhraseBank [11]) we used. We analyzed the FM Ratio in each dataset, there are higher FM Ratios in FiQA-headline and PhraseBank, but a lower Ratio in FiQA-post. This

---

[10]https://sites.google.com/view/fiqa/home
[11]https://www.researchgate.net/publication/251231364_FinancialPhraseBank-v10

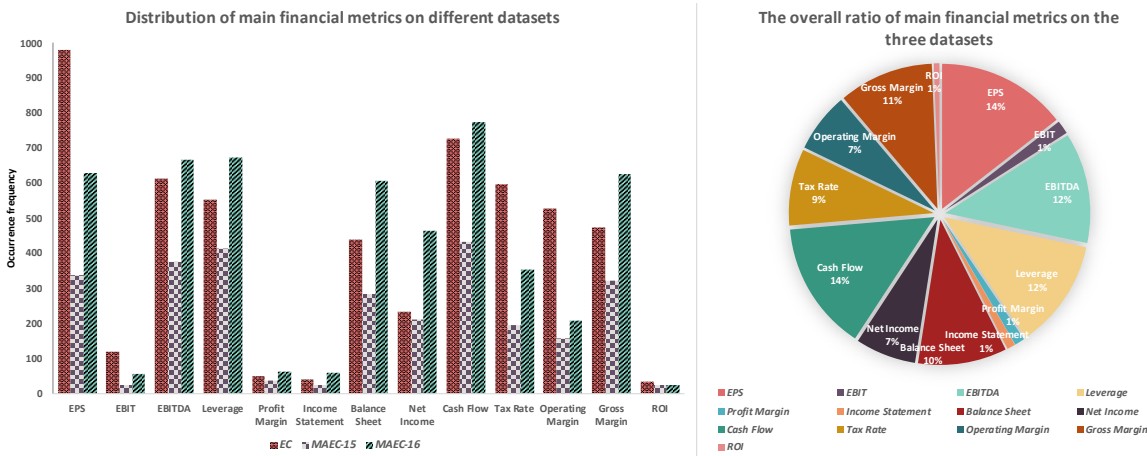

Figure 6: Occurrence frequency of main FMs on the three earnings call datasets.

Table 11: The impact of different text information on time series modeling. These cases are from the EC dataset, and the results are based on 3-day predictions.

| Case | Model | Top-8 Similar Points | Predictions | Absolute Difference | Ground Truth |
|---|---|---|---|---|---|
| American Tower Corp A (20171031) | BERT$_{BASE}$ | [0, 3, 7, 8, 11, 14, 22, 26] | -4.4576 | 0.1055 | -4.3521 |
| | KePt(w/o Knowledge) | [0, 3, 7, 8, 11, 14, 22, 26] | -4.4624 | 0.1103 | -4.3521 |
| | KePt | [0, 3, 7, 8, 11, 14, **18**, **22**] | **-4.4121** | **0.0600** | -4.3521 |
| Martin Marietta Materials (20171102) | BERT$_{BASE}$ | [1, 11, 15, 18, 21, 24, 25, 28] | -4.3885 | 0.1673 | -4.5558 |
| | KePt(w/o Knowledge) | [1, 11, **14**, 18, 21, 24, 25, 28] | -4.3939 | 0.1619 | -4.5558 |
| | KePt | [**4**, **14**, 15, 18, 21, 24, 25, 28] | **-4.4080** | **0.1478** | -4.5558 |
| Apache Corporation (20171102) | BERT$_{BASE}$ | [4, 9, 12, 15, 19, 24, 25, 28] | -4.1397 | 0.7434 | -3.3963 |
| | KePt(w/o Knowledge) | [4, 9, 12, 15, 19, 24, 25, 28] | -4.1442 | 0.7479 | -3.3963 |
| | KePt | [**2**, **7**, 12, 15, **16**, **19**, 25, 28] | **-4.0860** | **0.6897** | -3.3963 |

statistical result on FiQA-headline and FiQA-post is consistent with the effect of KePt on the two datasets, which is shown in Table 6.

We partitioned the dataset into training, validation, and test sets following a 7:1:2 ratio for each dataset. In the FiQA-headline and FiQA-post datasets, due to the unavailability of golden labels in the official test set, we partitioned 20% of the training set as a dedicated test set. In addition, We adopted the version of the dataset for which more than 50% agreement was reached as the comprehensive dataset for PhraseBank. This dataset was then partitioned according to the aforementioned ratios for training, validation, and testing.

### E.2 Experiment Settings

We fine-tuned each type of pre-trained model (BERT$_{BASE}$, KePt(w/o Knowledge) and KePT), along with an additional classification head, separately on the three datasets. The number of epochs for fine-tuning is set to 10, 10, and 5, respectively for FiQA-head, FiQA-post, and PhraseBank. The learning rate is 2e-5 by using the Adam optimizer with the default settings. The batch size is 32. We conduct experiments on the NVIDIA Tesla V100.

## F  Text Information and Time Series

Within the CTSP module, we introduced the Conditional Attention (CondAtt) module atop the Autoformer framework, aimed at incorporating text information. Within the CondAtt module, diverse text inputs yield varied impacts on time series modeling. Our analysis is presented in the Table 11. We delved into different text models (BERT$_{BASE}$, KePt (w/o Knowledge), and KePt) within CondAtt, and examined their conditional attention with historical time series information. We show the Top-8 data points of the time series according to text-to-time series attention weights (the overall length of the historical time series data point is 29) for analysis. We also provide the Ground Truth, predicted results (Predictions), and the absolute difference between the Ground Truth and Predictions (Absolute Difference) for each case. It is evident that distinct textual inputs capture diverse time series information, thereby influencing time-series predictions. It is worth noting that KePt's text input improves the modeling of time series information, resulting in Predictions that are closer to the Ground Truth and have a smaller Absolute Difference.