# OpenReview forum: "KeFVP: Knowledge-enhanced Financial Volatility Prediction"
_EMNLP/2023/Conference — EMNLP 2023 Findings_

### Official Review · Reviewer_8Q4Z · 2023-08-01

**Typos Grammar Style And Presentation Improvements:** In Figure 1
**Soundness:** 3

**Excitement:**

3: Ambivalent: It has merits (e.g., it reports state-of-the-art results, the idea is nice), but there are key weaknesses (e.g., it describes incremental work), and it can significantly benefit from another round of revision. However, I won't object to accepting it if my co-reviewers champion it.

**Missing References:**

1.	Araci, Dogu. "Finbert: Financial sentiment analysis with pre-trained language models." arXiv preprint arXiv:1908.10063 (2019).

2.	Raj Shah, Kunal Chawla, Dheeraj Eidnani, Agam Shah, Wendi Du, Sudheer Chava, Natraj Raman, Charese Smiley, Jiaao Chen, and Diyi Yang. 2022. When FLUE Meets FLANG: Benchmarks and Large Pretrained Language Model for Financial Domain. In Proceedings of the 2022 Conference on Empirical Methods in Natural Language Processing, pages 2322–2335, Abu Dhabi, United Arab Emirates. Association for Computational Linguistics.


Also, the paper ignores literature in financial journals (Journal of Finance, Journal of Financial Economics, and Review of Financial Studies) on volatility prediction while doing a good literature review of top NLP venues. It will be better to have a balanced literature overview while writing an interdisciplinary paper like this one. This is in context with a strong statement on line #83.


**Paper Topic And Main Contributions:**

The paper applies various natural language processing methods/techniques available for an important financial task of volatility prediction. For this task, the paper first introduces the Knowledge-enhanced Adaptive Pre-training module (KePt) which takes advantage and tries to put emphasis on the financial metrics (FMs). Later it performs a Knowledge-enhanced financial volatility prediction (KeFVP) task by combining embedding from KePt transformer with past volatility data. It achieves SOTA over the baselines by a small margin for the volatility prediction task.

**Questions For The Authors:**

Q: Why preferential masking of financial words was not performed?

**Reasons To Accept:**

The paper is well written in general. The paper attempt to enhance the NLP methods for an important task in the financial domain. For the same, it tries to incorporate price data with text and enhance the pre-trained model. It also provides ablation studies, and a case study to show the value addition coming from KePt and KeFVP.

**Reasons To Reject:**

Use of weak baselines 1: The paper uses the BERT-base-uncased model as a foundation and enhances it with the financial text. Some existing pre-trained language models (FinBERT, FLANG-BERT) do something similar. Pre-training corpus of the FLANG includes a detailed description of 8000+ financial words from Investopedia including all the financial metrics used here. It also uses the dictionary for preferential masking. It will be good to see FinBERT or FLANG-BERT as a baseline against KePt-BERT.

Use of weak baselines 2: A more suitable non-text baseline for the task can be added. For example, what will be the performance of a simple logistics regression model for volatility prediction with independent variables like CDS spread, market cap, book-to-market ratio, implied volatility from options, cost of capital, etc? The better question to ask will be what additional value text adds over the other financial variable available. The model like simple regression also provides interpretability over the transformer-based models. Also, variables like CDS and implied volatility are updated at a much higher frequency compared to the release of earnings calls once a quarter. Other models like GARCH can be also used as a baseline.

Dataset time period: EC dataset covers only 1 year of data while MAEC data covers 2015 to 2018. While the proposed model outperforms suggested baselines the period of study is not sufficient to make a strong conclusion. It also doesn’t include any market-wide high volatility periods like COVID-crisis or the 2008 financial crisis.

Performance improvement is very small over the baselines used. MSE of Voltage for EC data is 0.31 and KeFVP is 0.30. Which in my opinion is not a significant improvement given the small time period of study.

It is not a reason to reject, but the paper abuses the ACL formatting guidelines which doesn’t require a minimum font size for tables and figure. Some of the figures and tables in both the main paper and appendix (which doesn’t have a page limit) have font sizes of less than 5. In some cases maybe 2 or 3. The paper could easily move some information in the appendix and use a readable font size.

**Reproducibility:**

4: Could mostly reproduce the results, but there may be some variation because of sample variance or minor variations in their interpretation of the protocol or method.

**Reviewer Confidence:**

4: Quite sure. I tried to check the important points carefully. It's unlikely, though conceivable, that I missed something that should affect my ratings.

---

> ### Author Rebuttal · Authors · 2023-08-29
>
> #### Dear reviewer, thank you for your comments and affirmation of our method.
>
> ## __1: Use of weak baselines 1: FinBERT, FLANG-BERT baselines, and preferential masking__
>
> In regard to FinBERT, Table 3 in the manuscript presents the performance of FinBERT on the Volatility Prediction task. Additionally, based on the description in the FinBERT paper, FinBERT was pre-trained using the MLM approach on financial datasets. Consequently, the pretraining methodology of FinBERT aligns with that of KeP(w/o Knowledge), as outlined in Table 4 in our manuscript. Therefore, the pre-training approach of FinBERT can be inferred from the outcomes of KePt (w/o Knowledge).
>
> Moreover, we conducted experiments involving FLANG-BERT, designated as FVP (FLANG-BERT) in the following Table. Notably, it's important to acknowledge that the dataset used for training KePt-BERT is notably smaller than that of FinBERT and FLANG-BERT. The specific corpus sizes are provided in detail within this Table. This work primarily focuses on the task of Volatility Prediction. It is apparent that in this task, both FinBERT and FLANG-BERT exhibit less favorable performance compared to KePt-BERT.
>
> (Corpus Size of KeFVP and FVP(FinBERT) are counted by the number of sentences in each corpus, while FVP(FLANG-BERT) are counted by the number of documents in each corpus.)
>
> | Model                     | Corpus Size | MSE(Average) | MSE_3          | MSE_7          | MSE_15       | MSE_30       |
> |---------------------------|-------------|---------------|----------------|----------------|----------------|----------------|
> | FVP(BERT(base))            | -           | 0.318         | 0.644 (2.52e-2)| 0.311 (0.66e-2)| 0.196 (1.23e-2)| 0.120 (0.60e-2)|
> | FVP(FinBERT)              | 406,019     | 0.318         | 0.631 (2.22e-2)| 0.322 (0.72e-2)| 0.195 (1.19e-2)| 0.122 (0.74e-2)|
> | FVP(FLANG-BERT)           | 696,001     | 0.313         | 0.644 (5.75e-2)| 0.293 (1.86e-2)| 0.190 (0.78e-2)| 0.124 (0.59e-2)|
> | **FVP(KePt-BERT) (KeFVP)** | **8,732**       | **0.300**     | **0.610 (3.31e-2)**| **0.291 (1.33e-2)**| **0.183 (0.89e-2)**| **0.114 (0.63e-2)**|
>
> Furthermore, to ensure fairness, we adopted the preferential masking of financial words method from FLANG-BERT and applied it to our KePt dataset. This led to the training of the FLANG-BERT(KePt) model from the bert-base-uncased checkpoint. The corresponding results are exhibited in the following Table under the FLANG-BERT(KePt).
>
> (The results of KePt on financial sentiment analysis. KePt(w/o Knowledge) denotes adaptively pre-training on the KePt dataset without knowledge injection.)
>
> | **Model**               | **FiQA-headline**  | **FiQA-post** | **PhraseBank** |
> |                      | MSE ↓         | R^2 ↑         | MSE ↓     | R^2 ↑     | Acc (%) ↑ | F1 (%) ↑  |
> |----------------------|---------------|---------------|-----------|-----------|-----------|-----------|
> | BERT(base)            | 0.115         | 0.297         | 0.101     | 0.342     | 80.54     | 82.97     |
> | KePt(w/o Knowledge)  | 0.100         | 0.297         | 0.084     | 0.438     | 78.40     | 82.56     |
> | FLANG-BERT(KePT)     | 0.091         | 0.396         | 0.100     | 0.393     | 81.61     | **85.04**     |
> | KePt                 | **0.080**     | **0.435**     | **0.077** | **0.488** | **82.73** | 83.80     |
>
>
> It is evident that for the Volatility Prediction task, even with a relatively smaller volume of data used for fine-tuning, the performance of KePt-BERT remains superior to that of FinBERT and FLANG-BERT. Furthermore, in the context of financial sentiment analysis, KePt outperforms BERT, FinBERT, and FLANG-BERT in general under the same training corpus (KePt dataset). This observation underscores the effectiveness of KePt.
>
> As described in the FLANG paper, FLANG employs over 8000 financial words and utilizes preferential masking of financial words during model training. However, FLANG does not incorporate any knowledge descriptions about these financial words. In contrast, our approach, KePt, leverages a subset of 253 financial metric (FM) vocabulary along with corresponding descriptions, intentionally excluding commonly used terms in general language to prevent potential semantic confusion caused by the ubiquitous lexicon. This selection is aimed at maintaining the specificity of financial language in the context of our method.
>
>
> ## __2: Use of weak baselines 2: Non-text Baselines__
>
> Furthermore, we have extended our experiments to include non-text baselines, namely the classical linear regression, GARCH, and ARIMA models. These additional baselines are presented in the following Table. Due to data availability constraints and the need to ensure comparability, we have maintained the use of historical price information as non-textual data, consistent with previous works. Notably, we have refrained from incorporating textual data from earnings calls in this context.
>
> | Model                           | MSE(Average) | MSE_3          | MSE_7          | MSE_15       | MSE_30       |
> |---------------------------------|---------------|----------------|----------------|----------------|----------------|
> | **LinearRegression**            | 0.622         | 0.995 (3.20e-2)| 0.595 (5.83e-2)| 0.495 (3.90e-2)| 0.402 (3.80e-2)|
> | **GARCH**                       | 1.756         | 2.084 (1.88)   | 1.729 (1.78)   | 1.657 (1.85)   | 1.555 (1.83)   |
> | **ARIMA**                       | 0.611         | 0.944 (6.94e-1)| 0.596 (6.66e-1)| 0.497 (7.10e-1)| 0.407 (5.78e-2)|
> | **KeFVP(w/o Text)**                 | 0.331         | 0.647 (2.55e-2)| 0.328 (1.52e-2)| 0.210 (1.32e-2)| 0.138 (1.35e-2)|
> | **KeFVP**                       | **0.300**     | **0.610 (3.31e-2)**| **0.291 (1.33e-2)**| **0.183 (0.89e-2)**| **0.114 (0.63e-2)**|
>
>
> We also agree with the potential benefits of leveraging variables such as CDS spread, implied volatility from option data, and financial metrics like market cap, book-to-market ratio, and cost of capital for enhancing Volatility Prediction, but we have deferred exploration of these variables to future works. This decision is consistent with the main focus of this paper on mining earnings call text, as well as the integration of text and time series data.
>
> Regarding the interpretability of transformer-based models, regression models offer intuitive explanations due to their fewer parameters. However, their limited fitting capacity can restrict their effectiveness. Transformer-based models exhibit robust fitting capabilities, albeit with a larger parameter count. These models can be explained visually through methods like attention visualization to some extent, which aids in the intuitive understanding of a substantial number of parameters and thereby contributes to result comprehension.
>
> Simultaneously, we introduced KeFVP (w/o Text) as a point of comparison. This outcome represents the results derived solely from the Autoformer model's treatment of time series data. In comparison to KeFVP, it is evident that the incorporation of textual information yields a significant enhancement in predictive performance compared with dealing with time series data in isolation.
>
> ## __3: Dataset Time Period__
>
> Since our work follows the precedent of using the EC and MAEC datasets, all experiments are conducted within the scope of these two types of datasets. Examining high volatility periods such as the COVID-19 crisis or the 2008 financial crisis presents a more challenging research problem, but no one in the community has compiled relevant datasets for algorithm research. With your prompt, we also have an algorithm for this problem research is of great interest. We will sort out the data during the relevant high volatility period and conduct corresponding algorithm research in future work. In fact, the data sorting work is already in progress.
>
> ## __4: Performance Improvement__
>
> Relative to VolTAGE, KeFVP employs a narrower set of data types. While VolTAGE utilizes earnings call text, historical price data, audio data from earnings calls, and a graph constructed from inter-company relationships within the dataset, KeFVP solely relies on textual and price data. Notably, despite this narrower scope, KeFVP achieves superior performance compared to VolTAGE.
>
> It's interesting to note that following VolTAGE's approach by introducing audio and graph data could potentially lead to further performance improvements in KeFVP. Integrating audio and graph data might provide additional context and insights that could enhance the model's predictive capabilities. This is indeed a promising avenue for future work and development.
>
>
> ## __5: Other Concerns__
>
> * Font Size: We will adjust the font size in the camera-ready version to enhance readability.
>
> * Reference: In fact, we have cited the work (Araci, Dogu. 'FinBERT: Financial sentiment analysis with pre-trained language models.' arXiv preprint arXiv:1908.10063 (2019)) in the paper. In addition, we also cite this important related work (Raj Shah, Kunal Chawla, Dheeraj Eidnani, Agam Shah, Wendi Du, Sudheer Chava, Natraj Raman, Charese Smiley, Jiaao Chen, and Diyi Yang. 2022. 'When FLUE Meets FLANG: Benchmarks and Large Pretrained Language Model for Financial Domain.' In Proceedings of the 2022 Conference on Empirical Methods in Natural Language Processing, pages 2322–2335, Abu Dhabi, United Arab Emirates. Association for Computational Linguistics) in the camera-ready version.
>
> Following your guidance, we intend to enrich the camera-ready version by including references from financial journals pertaining to Volatility Prediction. This addition aims to provide a comprehensive literature overview.
>
> * Typos Grammar Style And Presentation Improvements: We will revise the presentation in Figure 1 to align with the descriptions in the text. Any additional issues related to phrasing, typos, or other concerns will also be addressed in the camera-ready version.

---

### Official Review · Reviewer_HZuA · 2023-08-02

**Soundness:** 4

**Excitement:**

3: Ambivalent: It has merits (e.g., it reports state-of-the-art results, the idea is nice), but there are key weaknesses (e.g., it describes incremental work), and it can significantly benefit from another round of revision. However, I won't object to accepting it if my co-reviewers champion it.

**Paper Topic And Main Contributions:**

Topic:

Knowledge-enhanced Financial Volatility Prediction. This paper proposes a knowledge-enhanced financial volatility prediction method (KeFVP) to inject knowledge of financial metrics into text comprehension by knowledge-enhanced adaptive pre-training (KePt) and effectively integrating text and price information by introducing a conditional time series prediction module.

Contribution:

This paper develops a knowledge-enhanced adaptive pre-training (KePt) method to inject FMs knowledge into PLMs and construct a specific KePt dataset centered on FMs for adaptive pre-training.

This paper proposes an FVP model, equipped with IE and CTSP modules, designed to amalgamate price and text information effectively.

This paper performs evaluations using three real-world earnings call datasets, and the results establish new state-of-the-art (SOTA) benchmarks.

**Questions For The Authors:**

Question 1: Can you explain the motivation of the Information Enhancement module?

Question 2: Both Autoformer Decoder(AutoDec) and Conditional Attention(CondAtt) capture the interaction between text and historical volatility information. What’s the differences? Why use these two components in conditional decoder?

Question 3:

Line 489, in Figure 2, the light gray vertical dashed line should be Day 28;

Line 536, not 0.161, from Table 4, FiQA-post on MSE (0.084-0.077=0.007);

Is that right?

**Reasons To Accept:**

1. Inject financial metrics knowledge into a pre-trained language model.

2. Design a model to combine price and text information for financial volatility prediction.

**Reasons To Reject:**

1. Injecting knowledge into a pre-trained language model is similar to ERNIE（Enhanced Language Representation with Informative Entities ）.

2. The Conditional Time Series Prediction module（bottom(b) of figure 1） seems a little complicated.

**Reproducibility:**

4: Could mostly reproduce the results, but there may be some variation because of sample variance or minor variations in their interpretation of the protocol or method.

**Reviewer Confidence:**

2: Willing to defend my evaluation, but it is fairly likely that I missed some details, didn't understand some central points, or can't be sure about the novelty of the work.

---

> ### Author Rebuttal · Authors · 2023-08-29
>
> Dear reviewer, thank you for your comments and affirmation of our method.
>
> ## __1: Relationship with ERNIE__
>
> We have identified two distinct aspects of our approach:
>
> ### Divergence in Motivation from ERNIE:
> Our motivation diverges from ERNIE's focus. We aim to enhance Volatility Prediction by enabling pre-trained models to better comprehend the content of earnings calls. During our investigation, we observed that over 50\% of sentences in earnings calls encompass financial metric words (FM), such as EBIT and EPS, as shown in Table 1 in our manuscript. These FMs hold paramount importance in understanding a company's performance. Hence, our strategy involves incorporating detailed descriptions pertaining to these FMs as knowledge. This augmentation seeks to elevate the effect of models in Volatility Prediction tasks based on earnings call text.
>
> ### Incorporating Sentence-level Information:
> The FM knowledge we incorporate consists of elaborate descriptions rather than spans, in contrast to ERNIE. These descriptions are full sentences. Our methodology indirectly validates the significance of incorporating sentence-level information. This is also effective in improving the performance of the model, which makes it possible to add more information in our way.
>
> ## __2: About Conditional Time Series Prediction (CTSP) module and Question 2__
>
> Within the CTSP module, we introduced the Conditional Attention (CondAtt) module atop the Autoformer framework, aimed at incorporating text information. CondAtt is designed to enable textual data to influence time-series predictions. Unlike the Autoformer decoder (AutoDec) (consistent with the Autoformer ), which only captures time series information, the addition of CondAtt enables textual information to influence time series modeling. Our experiments involving CondAtt are shown in the table below.
>
> In addition to the Autoformer model, we extended the implementation of CondAtt to various other transformer-based time series models (CTFM = Conditional Transformer, CInFM = Conditional Informer, CFedFM = Conditional FEDformer, CAtFM = Conditional Autoformer). DeFVP(Y w/o CondAtt) denotes the variant without CondAtt(·). We conducted ablation studies on the EC dataset, revealing that the inclusion of CondAtt significantly enhances the performance of these transformer-based models when combining textual information for improved time series modeling.
>
> * Informer: Haoyi Zhou, Shanghang Zhang, Jieqi Peng, Shuai Zhang, Jianxin Li, Hui Xiong,
> and Wancai Zhang. 2021. Informer: Beyond Efficient Transformer for Long 1034 Sequence Time-Series Forecasting. In Thirty-Fifth AAAI Conference on Artificial 1035 Intelligence, AAAI 2021.
>
> * FEDformer: Tian Zhou, Ziqing Ma, Qingsong Wen, Xue Wang, Liang Sun, and Rong Jin. 2022. 1039 FEDformer: Frequency Enhanced Decomposed Transformer for Long-term Series Forecasting. In International Conference on Machine Learning, ICML 2022.
>
> | Model                     | MSE(Average) | MSE_3          | MSE_7          | MSE_15       | MSE_30       |
> |---------------------------|---------------|----------------|----------------|----------------|----------------|
> | DeFVP(CTFM w/o CondAtt)   | 0.845         | 1.184 (1.88e-1)| 0.829 (1.69e-1)| 0.722 (1.87e-1)| 0.646 (1.86e-1)|
> | **DeFVP(CTFM)**            | 0.381         | **0.725 (3.66e-2)**| **0.369 (1.44e-2)**| **0.247 (1.14e-2)**| **0.182 (1.12e-2)**|
> |---------------------------|---------------|----------------|----------------|----------------|----------------|
> | DeFVP(CInFM w/o CondAtt)  | 0.989         | 1.334 (3.49e-1)| 0.953 (3.24e-1)| 0.840 (3.37e-1)| 0.830 (3.46e-1)|
> | **DeFVP(CInFM)**           | 0.369         | **0.705 (3.07e-2)**| **0.351 (1.71e-2)**| **0.240 (1.03e-2)**| **0.178 (1.31e-2)**|
> |---------------------------|---------------|----------------|----------------|----------------|----------------|
> | DeFVP(CFedFM w/o CondAtt) | 0.353         | 0.670 (8.09e-2)| 0.367 (4.26e-2)| 0.242 (5.30e-2)| 0.134 (0.93e-2)|
> | **DeFVP(CFedFM)**          | 0.311         | **0.610 (3.00e-2)**| **0.304 (1.92e-2)**| **0.202 (2.38e-2)**| **0.126 (0.89e-2)**|
> |---------------------------|---------------|----------------|----------------|----------------|----------------|
> | DeFVP(CAtFM w/o CondAtt)  | 0.331         | 0.647 (2.55e-2)| 0.328 (1.52e-2)| 0.210 (1.32e-2)| 0.138 (1.35e-2)|
> | **DeFVP(CAtFM)**           | **0.300**     | **0.610 (3.31e-2)**| **0.291 (1.33e-2)**| **0.183 (0.89e-2)**| **0.114 (0.63e-2)**|
>
>
> Moreover, within the CondAtt module, diverse text inputs yield varied impacts on time series modeling. Our analysis is presented in the following Table. We delved into different text models (BERT(base), KePt (w/o Knowledge), and KePt) within CondAtt, and examined their conditional attention with historical time series information. We show the Top-8 data points of the time series according to text-to-time series attention weights (the overall length of the historical time series data point is 29) for analysis. We also provide the Ground Truth, predicted results (Predictions), and the absolute difference between the Ground Truth and Predictions (Absolute Difference) for each case. It is evident that distinct textual inputs capture diverse time series information, thereby influencing time-series predictions. It is worth noting that KePt's text input improves the modeling of time series information, resulting in predictions (Predictions) that are closer to the Ground Truth and have a smaller Absolute Difference.
>
> We intend to incorporate these two sets of experiments into the camera-ready version of the paper.
>
> | Case                                    | Model               | Top-8 similar points            | Predictions | Absolute Difference | Ground Truth |
> |-----------------------------------------|---------------------|----------------------------------|-------------|--------------------|--------------|
> | American Tower Corp A (20171031)        | BERT(base)           | [0, 3, 7, 8, 11, 14, 22, 26]    | -4.4576     | 0.1055             | -4.3521      |
> |                                         | KePt(w/o Knowledge) | [0, 3, 7, 8, 11, 14, 22, 26]    | -4.4624     | 0.1103             |    -4.3521          |
> |                                         | KePt                | [0, 3, 7, 8, 11, 14, **18**, **22**]    | -4.4121     | **0.0600**             |      -4.3521        |
> |-----------------------------------------|---------------------|----------------------------------|-------------|--------------------|--------------|
> | Martin Marietta Materials (20171102)    | BERT(base)           | [1, 11, 15, 18, 21, 24, 25, 28] | -4.3885     | 0.1673             | -4.5558      |
> |                                         | KePt(w/o Knowledge) | [1, 11, **14**, 18, 21, 24, 25, 28] | -4.3939     | 0.1619             |      -4.5558        |
> |                                         | KePt                | [**4**, **14**, 15, 18, 21, 24, 25, 28] | -4.4080     | **0.1478**             |      -4.5558        |
> |-----------------------------------------|---------------------|----------------------------------|-------------|--------------------|--------------|
> | Apache Corporation (20171102)           | BERT(base)           | [4, 9, 12, 15, 19, 24, 25, 28]  | -4.1397     | 0.4734             | -3.3963      |
> |                                         | KePt(w/o Knowledge) | [4, 9, 12, 15, 19, 24, 25, 28]  | -4.1442     | 0.7479             |        -3.3963       |
> |                                         | KePt                | [**2**, **7**, 12, 15, **16**, **19**, 25, 28]  | -4.0860     | **0.6897**             |       -3.3963        |
>
> ## __3: About Information Enhancement (IE) module__
>
> As demonstrated in Table 1 (Statistics on datasets) in our manuscript, the datasets EC, MAEC-15, and MAEC-16 encompass a substantial number of sentences within their earnings call transcripts, averaging around 100 sentences per transcript. Consequently, prior to integrating textual information into time series prediction modules like CTSP, it becomes essential to further model the wealth of sentence-level information to capture crucial insights. In this regard, we introduce the Information Enhancement (IE) module. This module employs attention mechanisms at the sentence dimension. The objective is to distill the vital elements from the multitude of sentences in each transcript, thus better serving the CTSP module.
>
> In Table 3 of our manuscript, the ablation studies on the IE module are presented. The results underscore the significant role of the IE module in performance enhancement. In the camera-ready version, we intend to visualize the attention weights within the IE module, offering a more intuitive depiction of how the IE module aids in capturing key information from the numerous sentences.
>
> Results of ablation study on EC dataset. **KeFVP** denotes our overall method; KeFVP(w/o CTSP \& IE) denotes removing CTSP and IE, and only text remains; KeFVP(w/o IE) denotes removing information enhancement (IE).
> | Pattern | Model                 | MSE(Average) | MSE_3          | MSE_7          | MSE_15       | MSE_30       |
> |---------|-----------------------|---------------|----------------|----------------|----------------|----------------|
> | FVP     | KeFVP(w/o CTSP & IE)  | 0.392         | 0.743 (2.94e-2)| 0.380 (1.72e-2)| 0.258 (1.43e-2)| 0.185 (1.17e-2)|
> |         | KeFVP(w/o IE)         | 0.319         | 0.642 (4.80e-2)| 0.308 (2.08e-2)| 0.198 (1.60e-2)| 0.129 (1.38e-2)|
> |---------|-----------------------|---------------|----------------|----------------|----------------|----------------|
> |         | **KeFVP**              | **0.300**     | **0.610 (3.31e-2)**| **0.291 (1.33e-2)**| **0.183 (0.89e-2)**| **0.114 (0.63e-2)**|
>
>
> ## __4: Other Concerns__
>
> Thanks for pointing out these issues. I understand that minor oversights can happen during the writing process. We are planning to address these issues in the camera-ready version.

---

### Official Review · Reviewer_QcTY · 2023-08-04

**Soundness:** 4

**Excitement:**

4: Strong: This paper deepens the understanding of some phenomenon or lowers the barriers to an existing research direction.

**Paper Topic And Main Contributions:**

To address the challenges of financial volatility prediction tasks, this paper presents a novel method called FVP (Financial Volatility Prediction). In specific, the authors develop a method called KePt to enhance PLMs with financial features, such as EPS, EBIT ROI etc. Second, the FVP model considers historical price and text information. The experiments are conducted on three real-world datasets (Earings Call Data) and show that the proposed solution has significant improvement compared with a number of SOTA benchmarks.

**Reasons To Accept:**

- The paper is well-written and easy to follow.
- The proposed solution is novel and sound, particularly, constructing the KePt dataset, which is useful to enhance PLMs for financial tasks and will make the dataset publicly available.
- The analysis is comprehensive, including ablation study and case study.



**Reasons To Reject:**

- A few parts lack details. E.g., Line 220, we sift through the financial corpora to isolate sentences that include these metrics. (how to identify these metrics).

- It is good to run the experiments multiple times and report the mean and std.

**Reproducibility:**

4: Could mostly reproduce the results, but there may be some variation because of sample variance or minor variations in their interpretation of the protocol or method.

**Reviewer Confidence:**

4: Quite sure. I tried to check the important points carefully. It's unlikely, though conceivable, that I missed something that should affect my ratings.

---

> ### Author Rebuttal · Authors · 2023-08-29
>
> Dear reviewer, thank you so much for your detailed comments and affirmation of our approach.
>
> We have compiled a vocabulary of 253 Financial Metrics (FM) along with their corresponding knowledge descriptions. Utilizing Spacy's EntityRuler functionality, we performed Named Entity Recognition (NER) to identify FM terms within sentences. This process enabled us to filter sentences containing FM terms, thereby forming the KePt dataset. In the camera-ready version, we will augment this detail and provide a more detailed description of the dataset construction process.
>
> In fact, in the manuscript of this paper, we presented the average results (Mean(Std)) of 10 runs along with their corresponding standard deviations in Tables 2, 3, and 5.

---

### Meta-Review · Area_Chair_BVXn · 2023-09-23

**Recommendation:** 3

**Metareview:**

The reviewers agree that the authors present a new method for FVP and show SOTA performance on three benchmark datasets. The paper is generally well-written with comprehensive analysis. The authors did a good job in making a detailed response to the reviewer's queries and clarifying their doubts. The new study on non-text baselines w.r.t. the reviewer comments forms an important artifact to the paper and should be included in the subsequent versions. The same goes for the relevant references from non-NLP journals (financial) to provide more context in the literature review section for this type of interdisciplinary research. In light of these arguments, the paper has merit to feature in the conference.

---

### Decision · Program_Chairs · 2023-10-07

**Decision:**

Accept-Findings

**Comment:**

The reviewers agree that the authors present a new method for FVP and show SOTA performance on three benchmark datasets. The paper is generally well-written with comprehensive analysis. The authors did a good job in making a detailed response to the reviewer's queries and clarifying their doubts. The new study on non-text baselines w.r.t. the reviewer comments forms an important artifact to the paper and should be included in the subsequent versions. The same goes for the relevant references from non-NLP journals (financial) to provide more context in the literature review section for this type of interdisciplinary research. In light of these arguments, the paper has merit to feature in the conference.